

# Associations between picocyanobacterial ecotypes and cyanophage host genes across ocean basins and depth

Clara A. Fuchsman, David Garcia Prieto, Matthew D. Hays and Jacob A. Cram

Horn Point Laboratory, University of Maryland Center for Environmental Science, Cambridge, MD, United States of America

## ABSTRACT

**Background**. Cyanophages, viruses that infect cyanobacteria, are globally abundant in the ocean's euphotic zone and are a potentially important cause of mortality for marine picocyanobacteria. Viral host genes are thought to increase viral fitness by either increasing numbers of genes for synthesizing nucleotides for virus replication, or by mitigating direct stresses imposed by the environment. The encoding of host genes in viral genomes through horizontal gene transfer is a form of evolution that links viruses, hosts, and the environment. We previously examined depth profiles of the proportion of cyanophage containing various host genes in the Eastern Tropical North Pacific Oxygen Deficient Zone (ODZ) and at the subtropical North Atlantic (BATS). However, cyanophage host genes have not been previously examined in environmental depth profiles across the oceans.

**Methodology**. We examined geographical and depth distributions of picocyanobacterial ecotypes, cyanophage, and their viral-host genes across ocean basins including the North Atlantic, Mediterranean Sea, North Pacific, South Pacific, and Eastern Tropical North and South Pacific ODZs using phylogenetic metagenomic read placement. We determined the proportion of myo and podo-cyanophage containing a range of host genes by comparing to cyanophage single copy core gene terminase (*terL*). With this large dataset (22 stations), network analysis identified statistical links between 12 of the 14 cyanophage host genes examined here with their picocyanobacteria host ecotypes.

**Results**. Picyanobacterial ecotypes, and the composition and proportion of cyanophage host genes, shifted dramatically and predictably with depth. For most of the cyanophage host genes examined here, we found that the composition of host ecotypes predicted the proportion of viral host genes harbored by the cyanophage community. Terminase is too conserved to illuminate the myo-cyanophage community structure. Cyanophage *cobS* was present in almost all myo-cyanophage and did not vary in proportion with depth. We used the composition of *cobS* phylotypes to track changes in myo-cyanophage composition.

**Conclusions**. Picocyanobacteria ecotypes shift with changes in light, temperature, and oxygen and many common cyanophage host genes shift concomitantly. However, cyanophage phosphate transporter gene *pstS* appeared to instead vary with ocean basin and was most abundant in low phosphate regions. Abundances of cyanophage host genes related to nutrient acquisition may diverge from host ecotype constraints as the same host can live in varying nutrient concentrations. Myo-cyanophage community in the anoxic ODZ had reduced diversity. By comparison to the oxic ocean, we can see which cyanophage host genes are especially abundant (*nirA, nirC,* and *purS*) or not

Corresponding author
Clara A. Fuchsman,
cfuchsman@umces.edu

abundant (myo *psbA*) in ODZs, highlighting both the stability of conditions in the ODZ and the importance of nitrite as an N source to ODZ endemic LLV *Prochlorococcus*.

## INTRODUCTION

Viruses are the most abundant biological entity in the ocean and are important players in biogeochemical cycling (*Fuhrman, 1999*). Viruses interface with their host's metabolism during infection and, therefore, are subject to environmental constraints imposed on their host (*Kelly et al., 2013*; *Motegi et al., 2015*). Most cellular transcription and translation stop under viral infection (*Doron et al., 2016*; *Waldbauer et al., 2019*), but the host provides the energy and resources for viral production. Many marine viruses encode genes for host function originally obtained from the host, known as viral host genes or auxiliary metabolic genes (*Mann et al., 2003*; *Roux et al., 2016*; *Moniruzzaman et al., 2020*). The virus uses these viral host genes to supplement key short-lived host proteins (*Lindell et al., 2005*; *Fridman et al., 2017*). Viral host genes are thought to increase viral fitness by synthesizing nucleotides for virus replication (*Lindell et al., 2007*; *Weigele et al., 2007*), enabling their hosts to acquire limiting nutrients (*Breitbart et al., 2007*; *Kelly et al., 2013*), alleviating energy limitation (*Thompson et al., 2011*; *Mahmoudabadi, Milo & Phillips, 2017*) or by otherwise dealing with the direct stress imposed by the environment (*Lindell et al., 2004*). The purpose of these genes is to help the virus produce progeny (*Lindell et al., 2004*; *Lindell et al., 2005*; *Lindell et al., 2007*). However, in failed or lysogenic infections, viral host genes from one host strain may provide new abilities to a different host strain (*Llorens-Marès et al., 2017*; *Popa, Landan & Dagan, 2017*; *Needham et al., 2019*). Though viral host genes can be important to viral success, they also have a cost because more genes increase the number of nucleotides required during replication and the N and P required for those nucleotides. Host-derived genes that are critical only in a specific environment, such as the surface ocean, will not provide fitness benefits in the deep euphotic zone, and therefore are modeled to not be encoded there (*Bragg & Chisholm, 2008*; *Hellweger, 2009*). Data from cyanophage host genes in the Eastern Tropical North Pacific (ETNP) and Bermuda Ocean Time Series (BATS) supports these models (*Fuchsman et al., 2021*). Therefore, the presence of a viral host gene suggests that a process is beneficial to the survival of a microbial viral factory between the time of infection and lysis and that these genes have been selected for under evolutionary pressure.

Picocyanobacteria and their viruses, cyanophages, are an ideal study system for viral host genes. Cyanophages are abundant and potentially important causes of mortality for the marine picocyanobacteria *Prochlorococcus* and *Synechococcus* (*Sullivan, Waterbury & Chisholm, 2003*; *Parsons et al., 2012*; *Baran et al., 2018*; *Carlson et al., 2022*). Picocyanobacteria are numerically dominant and contribute greatly to primary production (~40%) in offshore oligotrophic regions (*Rii, Karl & Church, 2016*). *Prochlorococcus*

and *Synechococcus,* are abundant under different ocean regimes, with *Prochlorococcus* dominating in the oligotrophic open ocean from 40°N to 40°S and the geographic extent of *Synechococcus* reaching both polar regions and high nutrient regimes (*Flombaum et al., 2013*). These photosynthesizers are at the base of the food chain and face mortality by grazing or viral infection.

Cyanophage are the best studied marine viruses. Picocyanobacterial infection levels can be low (1%) under stable conditions, but reach up to 10% in transition zones between ocean regimes (*Mruwat et al., 2021*; *Carlson et al., 2022*). Known cyanophage are all in the Caudovirales, the tailed bacteriophages (*Ackerman, 1998*) but belong to several families: the T4-like Myoviridae, T7-like Podoviridae, Siphoviridae, and unclassified BAC21E04 (*Mann et al., 2003*; *Sullivan, Waterbury & Chisholm, 2003*; *Sullivan et al., 2005*; *Sullivan et al., 2009*; *Huang et al., 2012*; *Labrie et al., 2013*; *Mizuno et al., 2013*; *Crummett et al., 2016*; *Flores-Uribe et al., 2019*). Myo-cyanophage are larger and usually more generalist while podo-cyanophage are smaller and more specific in their hosts (*Zborowsky & Lindell, 2019*). Each family of cyanophages encodes host-derived genes with the number and organization of these genes varying (*Labrie et al., 2013*; *Crummett et al., 2016*; *Marston & Martiny, 2016*). Viral host genes have been examined in the environment but without linking them to specific viral types or hosts (*Hurwitz, Hallam & Sullivan, 2013*; *Hurwitz, Brum & Sullivan, 2015*; *Roux et al., 2016*; *Coutinho et al., 2017*; *Gazitúa et al., 2021*; *Jurgensen et al., 2022*). We have previously examined variation in cyanophage-specific host gene abundance and composition with depth in the euphotic zone but only at two sites: the offshore ETNP ODZ and BATS (*Fuchsman et al., 2021*). Further data is needed because the composition of the picocyanobacterial hosts change greatly with depth, and their viruses and viral host genes likely change concomitantly.

The euphotic zone has strong gradients of light, nutrients, and, in some cases, oxygen, providing different environmental conditions for *Prochlorococcus* between surface waters and depth. *Prochlorococcus* ecotypes and abundances change dramatically over the span of meters in a depth profile (*Ahlgren, Rocap & Chisholm, 2006*; *Johnson et al., 2006*; *Zinser et al., 2007*). *Prochlorococcus* can be phylogenetically separated into ecotypes that are adapted to significantly different conditions including high light surface waters (HLI and HLII), medium light levels (LLI), and very low light deep euphotic waters (LLII, LLVII [originally NCI], LLIV), as well as warmer (HLII) and colder (HLI) temperature ranges, and oxic or anoxic waters (*Rocap et al., 2003*; *Ahlgren, Rocap & Chisholm, 2006*; *Zinser et al., 2007*; *Lavin et al., 2010*; *Malmstrom et al., 2010*). Distinct ecotypes of *Prochlorococcus* (LLV [also AMZ1], LLVI [also AMZ2], and AMZ3) live in Oxygen Deficient Zones (ODZs), water with <10 nM $O_2$, when blue light reaches the anoxic waters, with LLV being the dominant ecotype (*Goericke, Olson & Shalapyonok, 2000*; *Cepeda-Morales et al., 2009*; *Lavin et al., 2010*; *Ulloa et al., 2021*). *Synechococcus* has ecotypes adapted to different temperatures, nutrient availability, and iron availability (*Sohm et al., 2016*), but these variations tend to be between ocean regimes. How cyanophage host genes covary with cyanobacterial ecotype has not been previously examined on a global scale.

Environmental factors shape the distribution of cyanobacterial ecotypes, but the scientific community is just beginning to investigate the corresponding changes in the cyanophage

community and viral host-derived genes at the appropriate resolution in the euphotic zone (*Luo et al., 2020*; *Fuchsman et al., 2021*). To cover more of the global ocean, we here extend our analyses from the ETNP Oxygen Deficient Zone, and the oxic BATS station in the North Atlantic Subtropical Gyre (*Fuchsman et al., 2021*) to 20 other metagenomic depth profiles, dominated by *Prochlorococcus*, with 4 or more samples in the euphotic zone, including an East-West transect in the North Atlantic (Geotraces GA03), North Pacific subtropical gyre (HOT), South Pacific subtropical gyre (Geotraces GP13), Mediterranean and the Eastern Tropical South Pacific Oxygen Deficient Zone (ETSP ODZ) (Fig. 1). With this increased dataset, we examined changes in proportions of cyanophage encoding each host gene with depth, and used network analysis to statistically link cyanophage host genes, cyanobacterial ecotypes, and environmental parameters.

## METHODS

### Compilation of metagenomic data

Cellular metagenomes, from biological material larger than 0.2 μm; which included both bacteria/archaea and the viruses that were actively infecting them, were obtained as follows: the entire GA03 North Atlantic Subtropical Gyre transect sampled in November 2011 (*Biller et al., 2018*), four stations from the GP13 South Pacific transect in June 2011 (*Biller et al., 2018*), Hawaii Ocean Timeseries (HOT) cruises from May (HOT 272), August (HOT 275) and November (HOT278) of 2015 (*Luo et al., 2020*), the Mediterranean Sea in October 2015 (*Haro-Moreno et al., 2018*), the Eastern Tropical South Pacific Oxygen Deficient Zone (ETSP ODZ) Station 9 from July 2013 (*Fuchsman, Cherubini & Hays, 2022*) and the Eastern Tropical North Pacific Oxygen Deficient Zone in April 2012 St 136 (*Fuchsman et al., 2017*). Most of the results from the Eastern Tropical North Pacific Oxygen Deficient Zone have already been published (*Fuchsman et al., 2021*). The Mediterranean samples were prefiltered through a 5 μm prefilter (*Haro-Moreno et al., 2018*). All other samples were not prefiltered. Station locations can be seen in Fig. 1 and coordinates can be found in Table S1.

BioGeotraces metagenomic sequences (*Biller et al., 2018*) were downloaded from GenBank bioproject PRJNA385854. CTD and nutrient data were downloaded from the British Oceanographic Data Centre (https://www.bodc.ac.uk/geotraces/) as part of the GEOTRACES 2021 Intermediate Data Product (IDP2021). The GA03 nitrite data shown here were published in *Jacquot & Moffett (2015)*. HOT metagenomes (*Luo et al., 2020*) were downloaded from Bioproject PRJNA352737. Nutrient and CTD measurements for these cruises can be found at Hawaii Ocean Time Series Data Organization and Graphical System (https://hahana.soest.hawaii.edu/hot/hot-dogs/) and in the original paper (*Luo et al., 2020*). Metagenomes from the Eastern Tropical South Pacific St 9 can be downloaded from Bioproject PRJNA704804. Hydrographic and nutrient data from this ETSP cruise are deposited at NODC as accession 0128141 and are previously published (*Peters et al., 2018*; *Fuchsman, Cherubini & Hays, 2022*). Metagenomes from the Eastern Tropical North Pacific can be downloaded from Bioproject PRJNA350692. Hydrographic and nutrient data from this ETNP cruise are deposited at the NODC as accession 0109846. Data for ETNP St
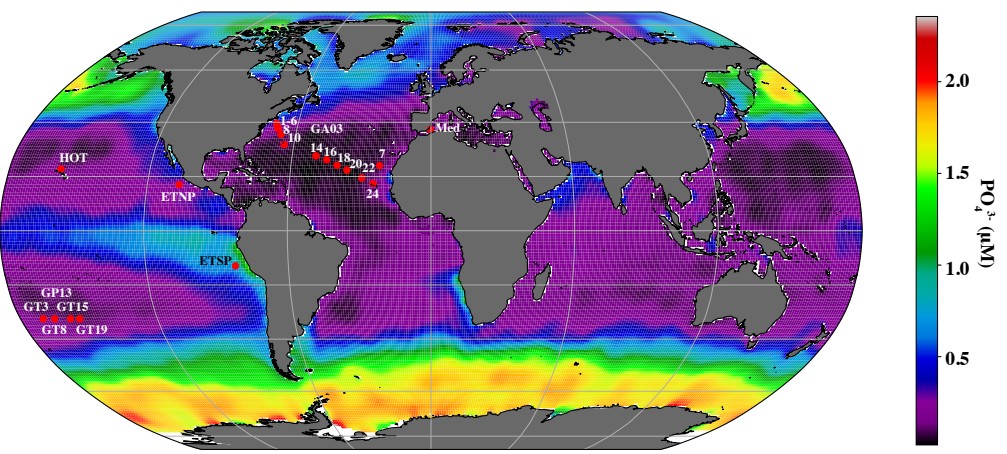

**Figure 1** **Map indicating stations examined in this study.** Map of stations with metagenomic depth profiles used in this study. Colors indicate surface phosphate concentrations from World Ocean Atlas 2018 (*Garcia et al., 2019*).

136 can be seen in *Fuchsman et al. (2017)*. Mediterranean metagenomes were downloaded from BioProject PRJNA352798, and CTD and nutrient information is contained in the paper (*Haro-Moreno et al., 2018*). Salinity data was estimated for the Mediterranean from World Ocean Atlas 2018 (*Zweng et al., 2019*). Ocean surface phosphate concentrations used in Fig. 1, were downloaded from World Ocean Atlas 2018 (*Garcia et al., 2019*). Metadata including latitude and longitude coordinates, mixed layer depth, temperature, salinity, oxygen and nutrient concentrations for all metagenomes can be found in Table S1. Mixed layer depths were determined by examining density profiles for each station, with the edge of the mixed layer determined by a change in density >0.03 kg/m$^3$.

## Phylogenetic trees and metagenomic read placement

Sixteen genes that are shared between cyanobacteria and cyanophage were examined (Table 1). Phylogenetic trees of 13 of these genes were adapted from *Fuchsman et al. (2021)*. These viral host genes were originally chosen based on their abundance in cyanophage assembled contigs from cellular metagenomes (*Fuchsman et al., 2021*), and these contigs were marked as viral using VirSorter (*Roux et al., 2015*). The cyanophage contigs were dominated by myo-cyanophage (*Fuchsman et al., 2021*), so our analysis here is subject to that bias. The trees were expanded by adding proteins from assembled contig sequences from BioGeotraces transects GA03 and GP13 (*Biller et al., 2018*). To find the genes of interest in the assembled contigs, genes were called and annotated with Prokka 1.14.6 (*Seemann, 2014*). A custom BLAST database (*Altschul et al., 1997*; BLAST v2.6.0+) was created using proteins from the assembled contigs from the two transects. Appropriate genes were obtained by blasting known reference sequences (blastp) against this database. The combined full-length sequences from the BioGeotraces transects and reference sequences from the pre-existing trees, which included ETNP assembled proteins, were aligned using

**Table 1   Cyanophage host gene names and functions.**

| Gene code | Protein name | Function |
|---|---|---|
| cobS | Cobalamin synthetase | Vitamin B12 synthesis |
| nirA | Assimilatory nitrite reductase | Reduce nitrite to ammonia for assimilation in biomass |
| nirC | FNT Nitrite transporter | Transports nitrite across the cytoplasmic membrane |
| psbA | Photosystem II protein D1 | Reaction core of Photosystem II |
| psbD | Photosystem II protein D2 | Reaction core of Photosystem II |
| speD | S-adenosylmethionine decarboyxylase | Polyamine biosynthesis |
| talC | Transaldolase | Pentose phosphate pathway |
| pbsA | Heme oxygenase | Iron acquisition? Phytobilin biosynthesis |
| thyX | Thymidylate synthase | Pyrimidine synthesis |
| pyrE | Orotate phosphoribosyltransferase | Pyrimidine synthesis |
| purN | Phosphoribosylglycinamide formyltransferase | Purine synthesis |
| purM | Phosphoribosylformyglcinamidine cycloligase | Purine synthesis |
| purC | Phosphoribosylaminoimidazole succinocarboxamide sythase | Purine synthesis |
| purS | Phosphoribosylformylglycinamidine synthase | Purine synthesis |
| pstS | High affinity phosphate binding protein | ABC Phosphate transporter from periplasm to cytoplasm |
| phoH | PhoH | Unknown |

MUSCLE v3.8.31 (*Edgar, 2004*). The alignment was used to construct an amino acid maximum likelihood phylogenetic tree using RaxML-ng v0.7.0 with tbe bootstrap analysis ($n = 100$) (*Kozlov et al., 2019*). Groups within the phylogenetic trees were then labeled based on the references within that group.

Some exceptions to the above pipeline apply. Cyanobacteria ecotypes were determined using the internal transcribed spacer (ITS) region of rRNA. This nucleotide tree was previously used (*Fuchsman et al., 2021*) and is originally from *Lavin et al. (2010)*, so ODZ group AMZ3 was not included. However, group AMZ3 was at very low abundances where it was found in the coastal ETSP as a single-cell (*Ulloa et al., 2021*). As noted previously (*Fuchsman et al., 2021*), the *psbA* tree was also in nucleotide space as the distinction between actual host genes and viral host genes is too small in amino acid space for this gene. The assimilatory nitrite reductase *nirA* phylogenetic tree was created using the cyanobacterial amino acid tree from *Widner et al. (2018)* and identified cyanophage contigs from *Gazitúa et al. (2021)* as well as assembled proteins from BioGeotraces (*Biller et al., 2018*). The *nirC* gene was identified in ETNP cyanophage contigs containing *nirA*. The *speD* and *nirC* phylogenetic trees were created de novo using BioGeotraces assembled proteins (*Biller et al., 2018*), ETNP assembled proteins (*Fuchsman et al., 2017*), and published reference genomes. All trees created for this publication can be found in the Supplemental Data File. The reference sequences used for these trees can be found at Figshare: https://doi.org/10.6084/m9.figshare.21899361.v1.

For phylogenetic read placement, first the reference sequences making up the tree were BLASTed (tblastn) very broadly (e-value $= 10^{-5}$) against an ETNP, ETSP, HOT, GP13, GA03, and Mediterranean metagenomic read databases. The short reads obtained were

then aligned to the reference tree using PaPaRa Parsimony-based Phylogeny-Aware Read Alignment program 2.0 (*Berger & Stamatakis, 2011*). Non-overlapping paired-end reads were then combined into one aligned sequence and placed on the tree by EPA-ng v0.3.5 with filter-max as 1 (*Barbera et al., 2019*). Placed reads have a pendant length indicating the similarity between a query read and the location it places on the tree. Reads that were placed with a pendant length greater than 2 were removed as previous work indicated they represent the wrong gene (*Fuchsman et al., 2017*). Less than 1% of reads were removed. The remaining reads were enumerated for each taxonomic group using the assign subcommand of Gappa v.0.4.0 and a taxonomy file listing the taxonomy of the tree reference sequences (*Czech, Barbera & Stamatakis, 2020*). Taxonomic read counts were normalized using the method previously described (*Fuchsman et al., 2019*) where normalization factors for each sample were determined by dividing the number of good quality reads in a sample by the 100 m ETNP sample. The read counts were multiplied by the sample normalization factor, divided by the length of the gene, and then multiplied by 100 to make visualization easier.

Single copy core genes were used in further analyses. Terminase large subunit (*terL*) was used as a core gene for the cyanophage groups, using the previously published tree (*Fuchsman et al., 2021*). Sipho-cyanophage included sequences from *Mizuno et al. (2013)* as well as *Sullivan et al. (2009)* and *Huang et al. (2012)*. Uncategorized BAC21E04 cyanophages (*Flores-Uribe et al., 2019*) were also examined, but not linked to viral host genes. To determine the proportion of myo or podo cyanophage containing a particular viral host gene, a ratio of cyanophage host gene/ cyanophage *terL* was obtained. These ratios were specific to each family of cyanophage. The single copy core gene for bacteria, RNA polymerase (*rpoB*) gene was used to determine the % of the bacterial and archaeal community that were picocyanobacteria. It was also used to determine the % of *Prochlorococcus* that had *nirA* and *nirC* genes.

## Statistics

We used *cobS* phylotypes to represent the myo-cyanophage community. Simpson's Index of Diversity (1-D) (*Simpson, 1949*) was calculated for *cobS* phylotypes using a python script, with a lower cut off of 0.8 normalized reads. An RDA plot of *cobS* phylotypes, as normalized reads, was created and fit to oxygen, depth, depth squared (to allow for nonlinear effects of depth) and ocean basin using the R package vegan v2.6.4; https://cran.r-project.org/web/packages/vegan/index.html. Analysis of variance (ANOVA), was calculated on each marginal term of the RDA output using the anova.cca function in the vegan package, to determine whether each of environmental variable was statistically related to the *cobS* community.

To explore the statistical interactions between viruses and cyanobacterial groups, we generated statistical association networks using a variation on the graphical lasso approach (*Tibshirani, 1996*). This approach used Lasso regression to identify which cyanobacteria and environmental parameters predicted the proportion of cyanophage with each viral host gene. However, our method differed from graphical lasso in that it did not explore associations between different viral genes nor between cyanobacterial groups.

To carry out this analysis, two matrices were generated. Matrix **X** contained the relative abundances of all cyanobacteria ecotype normalized reads, as well as the values of a suite of environmental and location parameters. Environmental parameters were: temperature, salinity, oxygen concentration, nitrate, nitrite, and phosphate. Location parameters were binary dummy variables corresponding to the ocean region in which the sample was collected. They included the following geographic regions: East and West and Central North Atlantic Ocean, Eastern Tropical North and South Pacific Oceans, Mediterranean Sea, and South Pacific Ocean. Location variables also indicated whether the sample was collected within the mixed layer, in a deep chlorophyll maximum, below the deep chlorophyll maximum, or in hypoxic ($<60\ \mu M\ O_2$), or oxygen deficient zone waters ($<10\ nM\ O_2$). Matrix **Y** contained the proportion of cyanophage with each viral host gene as determined by ratios with single copy core gene *terL*.

We used the R package glmnet (Version 4.1.4) to perform lasso regressions using all variables in the matrix **X** to predict each variable in **Y**. Lasso requires the specification of a tuning parameter, $\lambda$. We identified a single $\lambda$ value that optimized the predictive capability of the approach. To identify predictive capability, we used ten-fold cross-validation to minimize the residual mean squared error of our predictions. In this cross-validation approach, the data were split into ten even separate chunks. In an iterative process, each chunk was held out as test data, the model was trained on the remaining nine chunks. The performance of the model at predicting the test data was calculated by averaging the residual mean squared error (RMSE) of model predictions for each variable in **Y**, given the matrix **X** a given value of $\lambda$. We then used the 'optimize()' function in the R stats package (Version 3.6.3) to identify the $\lambda$ value associated with the lowest, cross-validated, combined RMSE. We recorded which variables were kept by the lasso approach, and by their relative penalized coefficients. We visualized the results of the lasso regressions using the R igraph package (Version 1.2.6) to show which viruses were statistically associated with each cyanobacterial host. We removed the environmental nodes from the visualization, but not the calculations, to allow the lasso to account for and factor out associations between viral genes and regions and environmental conditions. The code for creating these networks can be found at Figshare: https://doi.org/10.6084/m9.figshare.21498402.v1.

## RESULTS

Our dataset extended over a range of conditions, including both coastal and subtropical gyre stations, extremely low phosphate to replete phosphate surface waters (Fig. 1), undetectable to replete $NO_x^-$ (Figs. 2, 3, 4, 5 and 6; Figs. S1–S4, S6), and fully oxic to anoxic conditions (Fig. S7).

### Cyanobacteria ecotype and phage distributions

We used three genes to characterize the cyanobacteria and cyanophage across our dataset. For each station, we used the RNA polymerase *rpoB* gene to calculate the fraction of bacteria and archaea that are marine picocyanobacteria (% community). Within the picocyanobacteria, we use the ITS region of rRNA was used to categorize picocyanobacteria into ecotypes (Table S2). For *Prochlorococcus*, High Light (HL) I, High Light II, Low Light

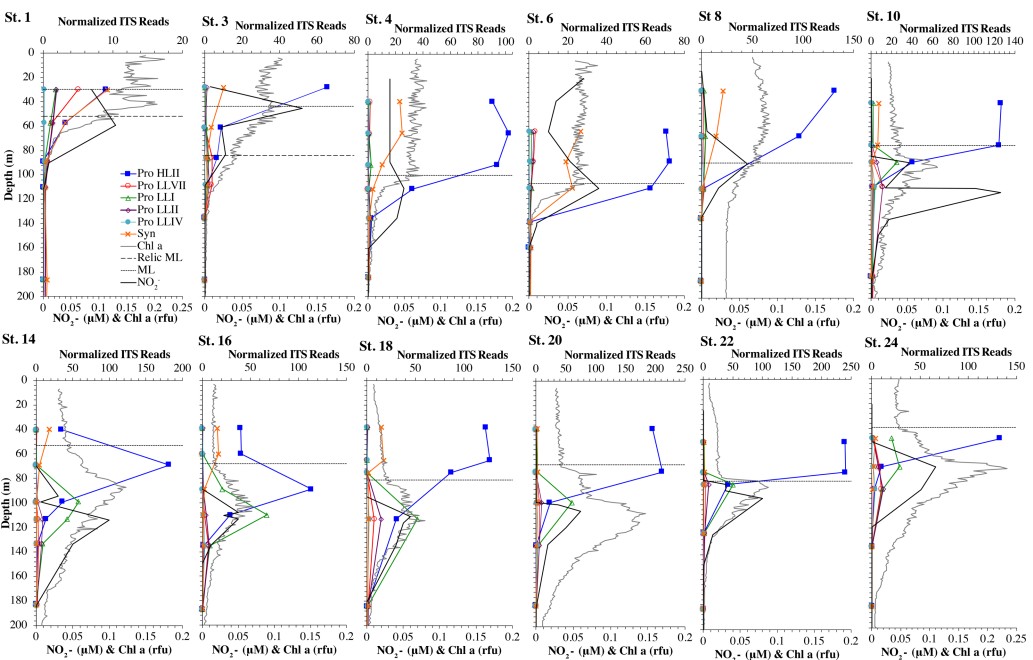

**Figure 2** **Cyanobacterial ecotypes for North Atlantic Geotraces transect GA03.** Cyanobacterial ecotypes as determined by internal transcribed spacer (ITS) normalized reads, chlorophyll, [NO$_2^-$] and mixed layer depth for the North Atlantic GA03 transect. The mixed layer (ML) is indicated by the dotted line and relic mixed layer (remnants of a previous mixed layer) is indicated by a dashed line. Nitrite concentrations are from *Jacquot & Moffett (2015)*.

(LL) I, Low Light II, Low Light VII which was originally called NCI, Low Light IV, Low Light V and Low Light VI ecotypes were examined. LLVI was never abundant. *Synechococcus* was only abundant in the North Atlantic GA03 transect and only in the Eastern part of this transect (section 1) (Fig. 2, Figs. S1–S2). The cyanophage single-copy core gene terminase large subunit (*terL*) was used to identify families of cyanophage in cellular genomes (Table S3). In general, myo-cyanophages were more abundant than podo-cyanophages in cellular metagenomes. However, the differences were sometimes slight. Sipho-cyanophages were always much less abundant than myo and podo-cyanophage, as were unclassified BAC21E04 cyanophages (Figs. 3–6, Figs. S1–S4).

Unclassified BAC21E04 cyanophages were only seen in some of the cellular metagenomes. BAC21E04 cyanophages had a maximum in the oxyclines at both the ETNP and ETSP, and were also present in the ODZ in the ETNP (Fig. 6). These unclassified cyanophages were present in GA03 section 1 (Fig. 3, Fig. S1) and in the Mediterranean (Fig. 5), but were barely detectable in any of the subtropical gyres (Figs. 3–4, Figs. S1–S4). This spotty representation is consistent with previous data (*Flores-Uribe et al., 2019*).

### North Atlantic

The GA03 transect occurred in November 2011. In the North Atlantic, November is part of the winter transition, where mixed layers are deepening (*Diaz et al., 2021*). The first section of the GA03 transect was across the continental slope (St 1, 3, 4, 6) to offshore

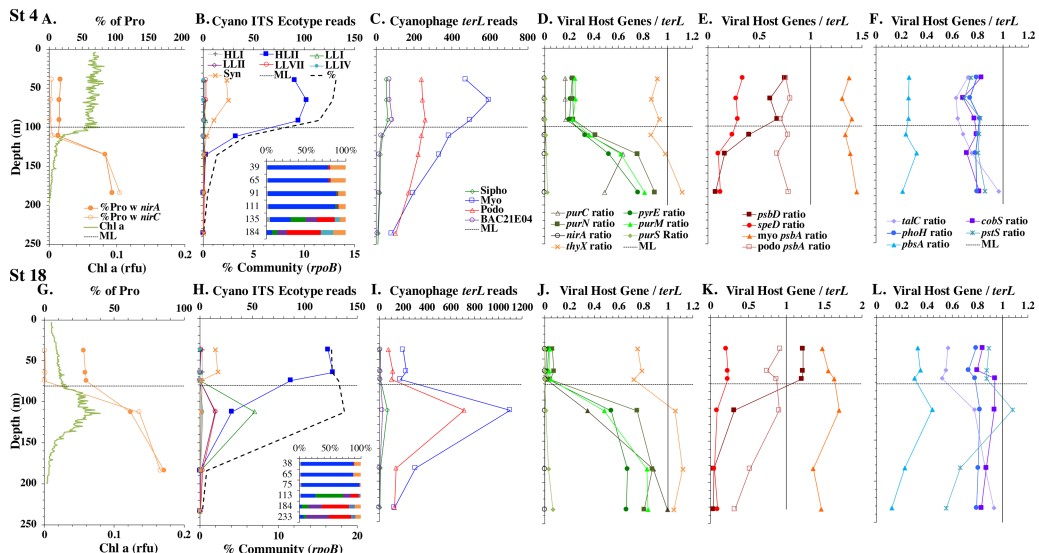

**Figure 3** Profiles of cyanobacteria, cyanophage, and cyanophage host genes for example North Atlantic stations. Example stations from GA03 transect section 1 (top) across the continental shelf and section 2 across the subtropical gyre (bottom). (A & G) The % of *Prochlorococcus* with nitrite assimilation genes *nirA* and *nirC*, and chlorophyll. (B & H) Cyanobacterial ITS ecotype normalized reads and percent of prokaryote community (RNA polymerase, *rpoB*; dashed bold line). Insert: Cyanobacterial ecotypes as % of Picocyanobacteria. Colors match the symbols. (C) & (I) Cyanophage abundance as determined by *terL* normalized reads. (D–F & J–L) The proportion of cyanophage with a host gene as measured by viral host gene/*terL* ratio. Full gene names and functions are seen in Table 1. The mixed layer (ML) is indicated by the dotted line.

waters (St 8) (*Lam, Ohnemus & Auro, 2015*), and the mixed layers or relic mixed layers in this section were quite deep (∼100 m) and there were no deep chlorophyll maxima (chlorophyll maxima below the mixed layer), but surface water temperatures were 23 °C (Fig. 2; Table S1). The picocyanobacteria made up a fairly constant fraction of the total microbial community throughout the mixed layer (10–15%) (Fig. 2, Fig. S1). HLII *Prochlorococcus* was the dominant ecotype throughout the water column with *Synechococcus* as the second most abundant cyanobacteria (Fig. 2). There were a variety of *Synechococcus* ecotypes including Clade II, Clade XV, Clade III, and Cyanobium (Fig. S5). In section 1 of the GA03 transect, the maximum of Low Light I *Prochlorococcus* ecotype below the mixed layer does not occur. At the deeper depths where the total numbers of picocyanobacteria were low, the low light ecotypes LLVII, LLII, and LLI were present (Fig. 2, Fig. S1).

The second section of the GA03 transect in late November and early December was a cross-basin section of the North Atlantic subtropical gyre (St 10, 14, 16, 18, 20, 22) (Fig. 1), but surface water temperatures were still 24−25 °C (Table S1). For these stations, the chlorophyll maxima and the maximum in picocyanobacteria (20–25% of the prokaryotic community) were both below the mixed layer at all sites (Fig. 2, Fig. S2). Nutrients were not detectable in the mixed layer with nitrite, nitrate and phosphate concentrations detectable at and below the chlorophyll maxima (Fig. 2, Fig. S6). HLII dominated in surface waters and at the picocyanobacteria maximum below the mixed layer (Figs. 2–3).

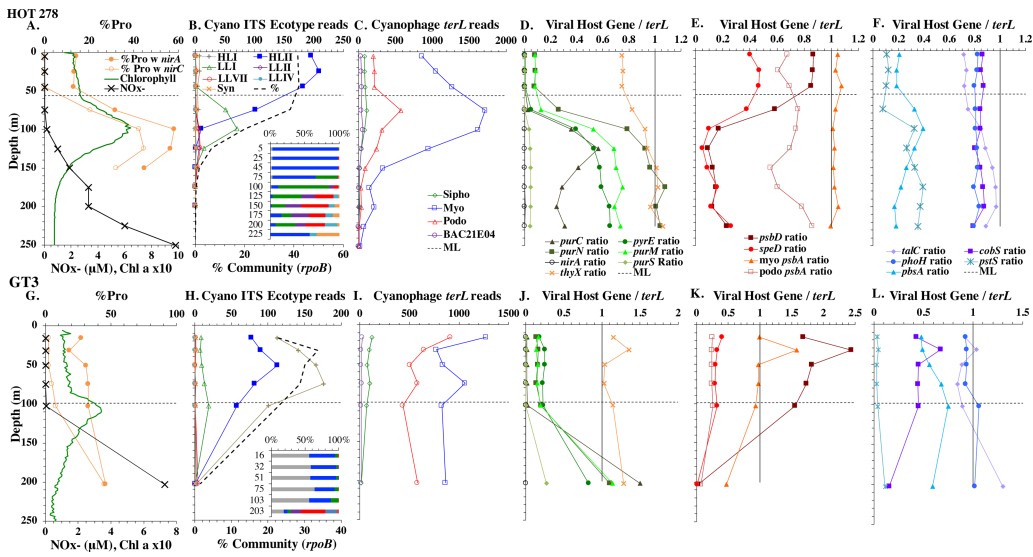

**Figure 4  Profiles of cyanobacteria, cyanophage, and cyanophage host genes for example Pacific stations.** Example dataset from the North Pacific subtropical gyre (HOT 278 from Nov 2015; A–F) and the South Pacific subtropical gyre (GP13 St GT3; G–L). Panels and axes are the same as for Fig. 3 except that panels (A & G) include $NO_x^-$ concentrations, and chlorophyll concentrations are multiplied by 10 to better fit on the axis. $NO_x^- = NO_3^- + NO_2^-$.

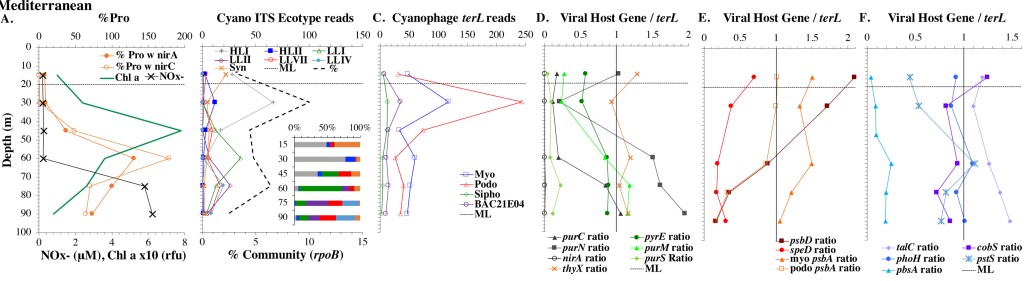

**Figure 5  (A–F) Profiles of cyanobacteria, cyanophage, and cyanophage host genes for the Mediterranean Sea.** Dataset from the Mediterranean Sea (Oct 2015). Panels and axes are the same as for Fig. 3 except that panel (A) includes $NO_x^-$ concentrations, and chlorophyll concentrations are multiplied by 10 to better fit on the axis. $NO_x^- = NO_3^- + NO_2^-$.

At the chlorophyll maxima, LLI had corresponding maxima, and then slightly deeper, LLII and LLVII had maxima (Fig. 2 and Fig. S2). *Synechococcus* were in low abundance for this section of the transect (0–2% of community) (Fig. S5). Myo-cyanophage were more abundant that podo-cyanophage in both GA03 sections (Fig. S2). Cyanophage abundance in cellular metagenomes often had its maximum either coincident or slightly deeper than the picocyanobacteria maximum (Fig. 2, Fig. S2). However, the in the subtropical gyre, the cyanophage maxima were wide, extending over ~100 m depth (Fig. S2).

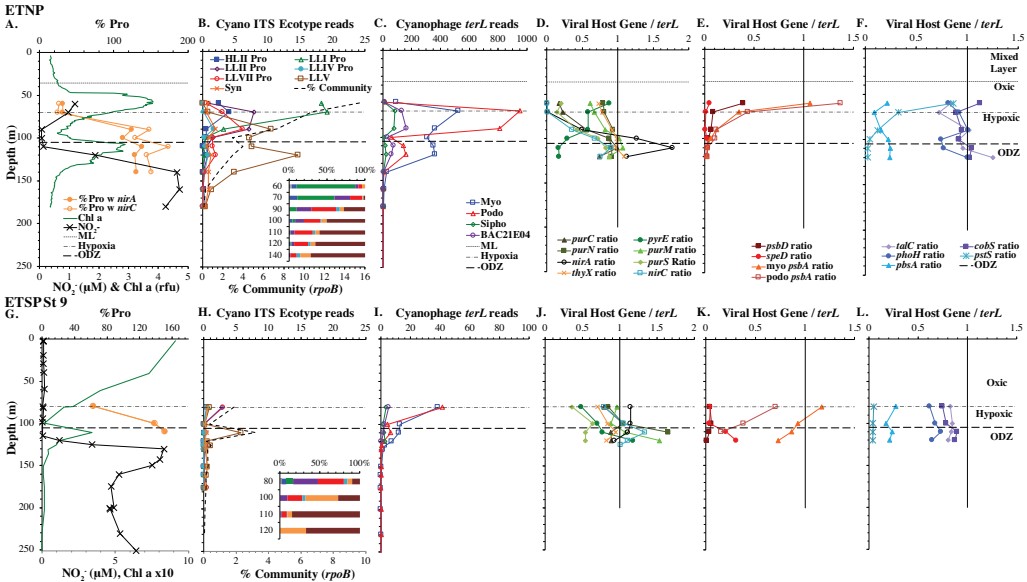

**Figure 6 Profiles of cyanobacteria, cyanophage, and cyanophage host genes for the Oxygen Deficient Zone stations.** Datasets from ETNP ODZ St 136 (A–F) and the ETSP ODZ St 9 (G–L). (A & G) The % of *Prochlorococcus* with nitrite assimilation genes *nirA* and *nirC*, NO$_2^-$ concentrations, and chlorophyll fluorescence. Fluorescence was multiplied by 10 in the ETSP but not in the ETNP. The rest of the panels are the same as found in Figs. 3–5, except that (D & J) include cyanophage nitrite transporter *nirC*. The mixed layer (ML) is indicated by the dotted line. The start of hypoxia (<60 μM O$_2$) is indicated by the dash-dot line. The top of the ODZ is the dashed line. Oxygen concentrations can be seen in Fig. S7.

### North Pacific subtropical gyre: Hawaii Ocean Timeseries

The depth resolution was particularly fine in the Hawaii Ocean Timeseries (HOT) 2015 datasets and covered the entire euphotic zone. We analyzed three time-periods at HOT (May, August, November) with a surface water temperature range of 24−27 °C (Table S1), but did not see large changes in cyanobacterial ecotypes and viral host genes between time-periods. Similar to GA03, at HOT, the *Prochlorococcus* ecotypes had a HLII to LLIV cascade with depth. Nutrients were low but measurable in the mixed layer (Fig. 4, Fig. S4), and picocyanobacteria had a maximum of 30–45% of the community in the mixed layer but dropped significantly in abundance in the lower euphotic zone (Fig. 4, Fig. S4). *Synechococcus* was ∼1% of community in surface waters at HOT. Myo-cyanophage were also more abundant that podo-cyanophage in cellular metagenomes. Cyanophage had wide maxima below the mixed layer, and cyanophage reads extended slightly deeper than their host (Fig. 4, Fig. S4).

### South Pacific subtropical gyre

The South Pacific (GP13) transect occurred in June 2011, which is during austral winter. The stations examined here (GT3, GT8, GT15, GT19) were all in the South Pacific Subtropical Gyre (*Ellwood et al., 2018*). At these stations, the top 5 samples were obtained in the mixed layer, and only the samples from 200 m represent waters below the mixed layer. The Deep Chlorophyll maximum at this site was below the mixed layer at ∼120 m

(*Ellwood et al., 2018*) and was not sampled for metagenomes. Rather 200 m represented the very bottom of the euphotic zone. Nutrients were low but detectable in the mixed layer (Fig. 4 and Fig. S3) and it is likely the productivity was limited by iron (*Ellwood et al., 2018*). Picocyanobacteria varied between 20–30% of the community in the mixed layer (Fig. 4 and Fig. S3), and HLI was the most abundant ecotype though HLII was still abundant (Fig. 4 and Fig. S3). Temperatures in surface waters at these stations were ∼19 °C (Table S1). The coexistence of HLI and HLII at those temperatures is consistent with previous data (*Zinser et al., 2007*). At 200 m, LLVII was the most abundant ecotype (Fig. 4 and Fig. S3), but total picocyanobacteria were only ∼2% of community. Myo-cyanophage were also more abundant that podo-cyanophage in cellular metagenomes from the South Pacific (Fig. 4 and Fig. S3).

### Mediterranean Sea

For the Mediterranean, samples were obtained in October 2015 (autumn) and nitrate and ammonia were both measurable in the mixed layer (*Haro-Moreno et al., 2018*). Salinity was not reported for this cruise (*Haro-Moreno et al., 2018*), but salinity from the same area in September 2014 was 38 psu in the euphotic zone (*Zweng et al., 2019*). As has been seen previously (*Mella-Flores et al., 2011*), HLI was the dominant ecotype in upper euphotic zone (Fig. 5) even though surface temperature was 23 °C (Table S1), which was warm enough for HLII (*Zinser et al., 2007*). The reasons for HLI dominance in the Mediterranean are unknown (*Mella-Flores et al., 2011*). The Mediterranean dataset showed a nice cascade of *Prochlorococcus* ecotypes. HLI was the dominant ecotype in the mixed layer and at the chlorophyll maximum (45 m), then LLI had its maxima at 60 m and LLII, LLVII and LLIV had their maxima at 75 m. Once again, the fraction of the prokaryotic community that were picocyanobacteria had a maximum below the mixed layer, but picocyanobacteria only represented 10% of the community at the maximum (30 m) (Fig. 5). The cyanophage abundance maximum in cellular metagenomes was also at 30 m (Fig. 5). Interestingly, podo-cyanophage dominated over myo-cyanophage at 30 and 45 m (the chlorophyll maximum) (Fig. 5).

## Eastern Tropical Pacific oxygen deficient zones

The ODZ stations had two chlorophyll maxima. In the ETNP, the primary chlorophyll maximum in oxic waters below the mixed layer and a thick secondary chlorophyll maximum in the ODZ. The ETSP had maximal chlorophyll at the surface and a thin secondary chlorophyll maximum in the ODZ. Nitrite concentrations, an indicator of complete anoxia, were <1 µM at the top of the ETNP and ETSP ODZs but increased at the bottom of the secondary chlorophyll maximum (Fig. 6; *Fuchsman et al., 2017*). In both ODZs, phosphate concentrations were extremely high (2.8–3 µM; Fig. S7). Surface DNA samples were not obtained for the ETNP and ETSP, but the ecotype cascade was present at depth with a LLI maximum at the primary chlorophyll maximum, a LLII and LLVII maximum above the ODZ, and an ODZ ecotype LLV maxima in both ODZs (Fig. 6). Picocyanobacteria were 3.5% of the prokaryotic community in the ETSP ODZ and 4% in the ETNP ODZ (Fig. 6). The ETSP ODZ had a much higher fraction of *Synechococcus* than

did the ETNP, but the ecotype of this *Synechococcus* did not appear to be on our tree (reads placed to nodes). Podo-cyanophage were more dominant than myo-cyanophage in cellular metagenomes in the oxyclines or hypoxic waters right above the ODZs (Fig. 6, Fig. S7). However, myo-cyanophage dominated in the ODZs. The oxyclines were almost the only areas in our dataset where podo-cyanophage were more dominant in cellular metagenomes than myo-cyanophage, with the Mediterranean being the only other example (Figs. 5 and 6).

## Cyanophage host gene/terL depth profiles

Cyanophage host genes for purine nucleotide synthesis genes (*purN, purC, purM*), pyrimidine nucleotide synthesis genes (orate phosphoribosyltransferase; *pyrE* and thymidylate synthase; *thyX*), photosynthesis Photosystem II D1 and D2 protein genes (*psbA* and *psbD*), polyamine biosynthesis gene (polyamine aminopropyl transferase; *speD*), assimilatory nitrite reductase (*nirA*) and nitrite transporter (*nirC*) and pentose phosphate pathway enzyme transaldolase (*talC*), heme oxygenase (*pbsA*) and phosphate transporter (*pstS*) were examined in depth profiles across ocean basins. Thymidylate synthase (*thyX*) has myo, podo, and sipho types, but only the myo form was ever abundant in this dataset (Fig. S8). Transaldolase (*talC*) has myo and podo and an unclassified cyanophage types (Supplemental Data). Only 20–30% of podo cyanophage appear to have *talC*, but if the unclassified cyanophage are also podo, then 80–100% of podo cyanophage contain *talC* (Fig. S9). Given the ambiguity, we do not explore podo *talC* further. We also examined *cobS* and *phoH,* which are generally considered viral host genes, but were previously suggested to be myo-cyanophage core genes by their phylogenetic trees and consistent depth profiles (*Fuchsman et al., 2021*). Cyanophage genes were compared to single-copy core gene *terL* for myo-cyanophage or podo-cyanophage to obtain the proportions of cyanophage with each gene. All cyanophage host gene / *terL* ratios can be found in Table S4.

Some of the cyanophage genes examined did not change with depth. Cyanophage genes *cobS* and *phoH* both had ratios with *terL* that were constant with depth and near one copy per genome (Figs. 3–6, Fig. S1–S4). Transaldolase (*talC*), used in the pentose phosphate pathway which creates nucleotide synthesis precursors (*Jacobson, Callaghan & Amador-Noguez, 2021*), also usually had a proportion near one copy per genome (Figs. 3–6, Figs. S1–S4). It is known that cyanophage transaldolase is a functional transaldolase (*Thompson et al., 2011*), but it is unknown if cyanophage *cobS* has the same function in the infected host as does the original host gene (cobalamin synthesis) (*Fuchsman et al., 2021*). The function of *phoH* is unknown in both host and phage (*Kazakov et al., 2003*). Cyanophage host gene *speD* also had a consistent depth profile, but was consistently present at low levels (~0.2 copies per genome). These genes do not seem to vary between host ecotypes.

One set of cyanophage host genes consistently increased at depth. The proportion of myo-cyanophage with purine synthesis genes *purN, purM,* and *purC* increased at the bottom of the euphotic zone throughout the ocean. We also see the proportion of myo-cyanophage with pyrimidine synthesis genes *pyrE* and *thyX* increase with depth, though *pyrE* decreases with depth only in the ETNP (Figs. 3–6, Figs. S1–S4). These increases

correspond to changes in *Prochlorococcus* ecotypes to extremely low light adapted types (Figs. 3–6, Figs. S1–S4). These nucleotide synthesis genes all increased with depth even at the stations in the GA03 transect section 1 where HLII was the only abundant *Prochlorococcus* throughout the euphotic zone (Fig. S1). However, at the deeper depths in section 1, where picocyanobacteria were only 0.5–2% community, extremely low light ecotypes were actually a higher proportion of cyanobacteria (Fig. 3, Fig. S1). The cyanophage host genes that increase with depth likely only provide benefit for viral production at depth where light levels are low.

A second set of cyanophage host genes consistently decreased with depth. The proportion of myo-cyanophage with photosynthesis gene *psbD* decreased with depth throughout the ocean (Figs. 3–6, Figs. S1–S4). *psbD* reached two copies per myo-cyanophage genome in surface waters at stations where HLI was abundant (Figs. 4 and 5, Fig. S3). The proportion of podo-cyanophage with photosynthesis gene *psbA* decreased with depth except at the stations in the GA03 transect section 1 (Figs. 3–6, Fig. S1–S4). The cyanophage host genes that decrease with depth likely only provide benefit for viral production in surface waters where light levels are high.

A subset of cyanophage host genes were particular to the ODZs. The assimilatory nitrite reductase gene *nirA* was found on a myo-cyanophage contig in the ETSP ODZ (*Gazitúa et al., 2021*). Here we find nitrite transporter *nirC* on two ETNP contigs from the cyanophage section of the *nirA* tree (120mP_k141_666106, 70m_k141_623198; Bioproject PRJNA350692). In our dataset, both cyanophage *nirA* and *nirC* were only abundant in the ODZs (Figs. 3–6, Figs. S1–S4). Cyanophage *nirC* was only significant in the ODZs, as no reads were found at any other stations (Table S4). In the ETSP, *nirA* and *nirC* reached one copy per myo-cyanophage, but in the ETNP, while *nirC* was one copy per myo-cyanophage, cyanophage *nirA* reached two copies per myo-cyanophage, indicating that *nirA* may also be found in podo-cyanophage, even though there are no known podo-cyanophage on the tree (Fig. 6, Supplemental Data). Additionally, the purine synthesis gene *purS* was only abundant in myo-cyanophage in ODZs, where it reached one copy per myo-cyanophage. However, myo-cyanophage *purS* was present at the bottom of the euphotic zone at low levels at other oxic stations reaching 0.1−0.2 copies per myo-cyanophage (Figs. 3–6, Figs. S1–S4). On the other hand, the proportion of myo-cyanophage with Photosystem II protein gene *psbA* was constant with depth except in both ODZs, where it decreased significantly (Figs. 3–6, Figs. S1–S4). This indicates that *psbA* did not confer an advantage for viral production in ODZs, while nitrite assimilation genes did.

In contrast, the proportion of myo-cyanophage with the phosphate transporter, phosphate binding protein gene (*pstS*) appeared to vary by ocean basin, similar to data seen previously in *Kelly et al. (2013)*. Interestingly, while the proportion of cyanophage with *pstS* is high at some stations, such as on the North Atlantic GA03 transect and in the Mediterranean Sea, and low at other stations, such as the South Pacific subtropical gyre GP13 stations and North Pacific subtropical gyre (HOT), the proportion of cyanophage with *pstS* at a particular station does not appear to change with depth (Figs. 3–6, Figs. S1–S4). Myo-cyanophage increased *pstS* copies at locations with lower average phosphate concentrations. However, at locations where phosphate can be extremely limiting, *pstS* was

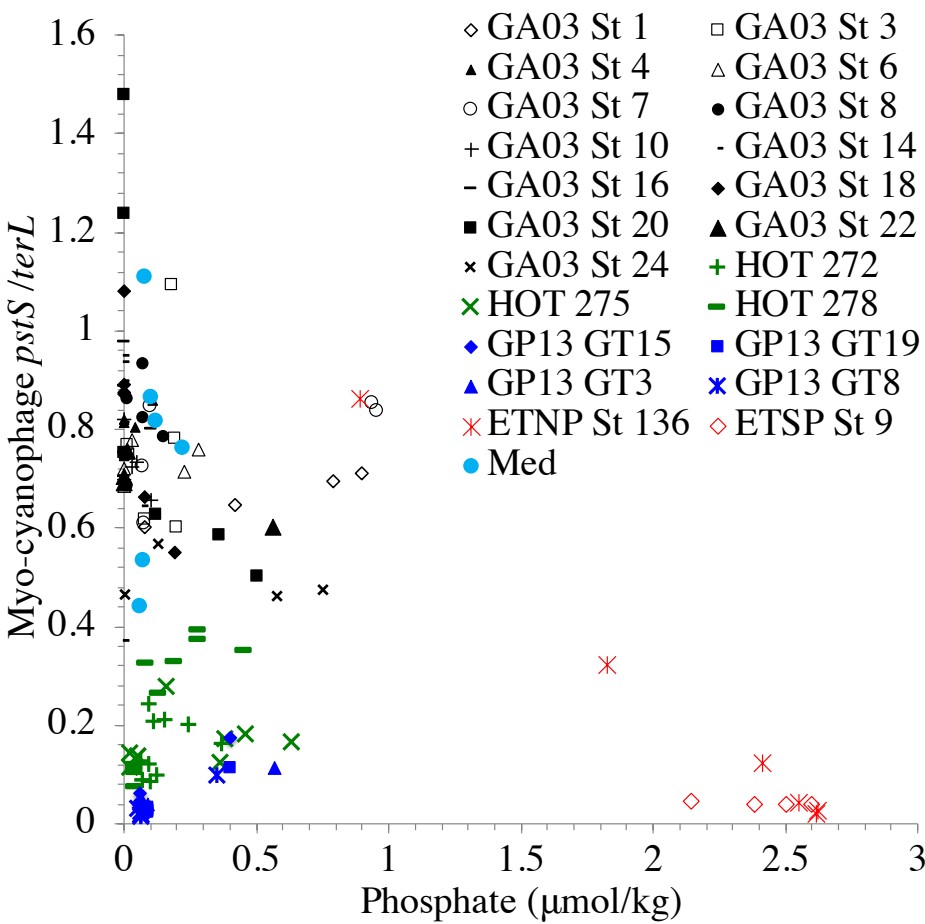

**Figure 7 Proportion of cyanophage with phosphate transporter *pstS* compared to phosphate concentrations.** Comparison of the proportion of myo-cyanophage with phosphate transporter gene *pstS versus* measured phosphate concentration across stations. GA03 samples are in black, HOT samples are in green, GP13 samples are in blue, Mediterranean samples are light blue and the ETNP and ETSP are in red.

retained in myo-cyanophage even if phosphate concentrations were higher in a particular sample (Fig. 7). These *pstS* profiles contrast with most of the other cyanophage host genes examined here because they are related to nutrient concentrations rather than host ecotype. Differing from oxic stations, in the ETNP, the proportion of cyanophage with *pstS* does change when switching from oxic to anoxic conditions; however phosphate concentrations also increase dramatically to extremely high concentrations (Fig. 7, Fig. S7; *Fuchsman et al., 2021*).

Cyanophage host gene heme oxygenase (*pbsA*) may also have varied across ocean basins. The proportions of myo-cyanophage with heme oxygenase gene (*pbsA*) were only ~0.2 in the Mediterranean, the GA03 shelf slope transect (section 1) and the ODZs. However, proportions reached 0.4 in the North Atlantic Gyre (GA03 section 2) and HOT. The highest proportions were seen at the South Pacific GP13 (~0.6) (Figs. 3–6, Figs. S1–S4). However, the abundances of this *pbsA* gene were low enough that patterns remain unclear.

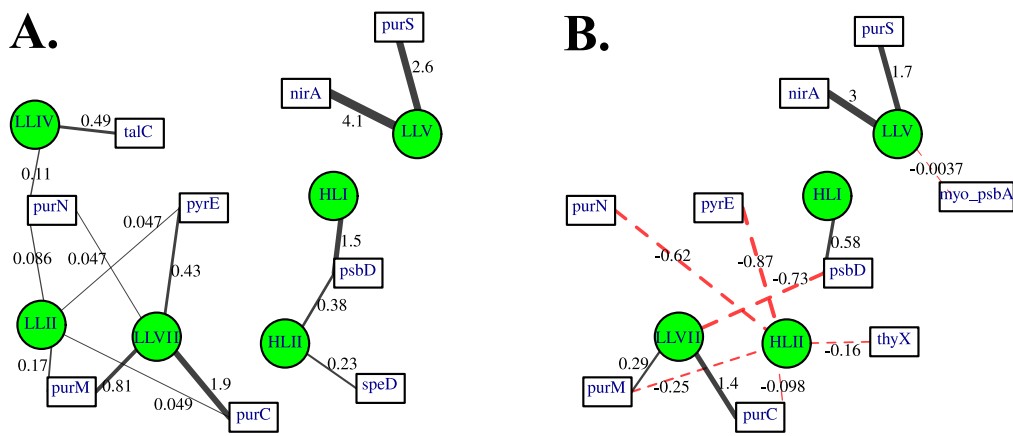

**Figure 8  A lasso-based association network between different cyanobacterial ecotypes and cyanophage host genes.** Nodes indicate the proportion of cyanophage with a viral gene (white squares), and cyanobacterial host ecotype abundances (green circles). Edges connect viral and cyanobacterial nodes that are statistically associated. Relative edge thickness relates to the relative coefficients of the Lasso regressions, with thicker edges associating with stronger statistical associations. The lasso regression coefficients are also shown as numbers. Environmental and location data were used to generate this network in order to factor out those variables, but those nodes and their associated edges are not shown to reduce complexity and to focus on the virus to host associations. (A) Only positive associations were used in the calculation, and (B) both positive (solid black edges) and negative associations (dashed red edges) were used.

## Statistical associations between viral host genes and cyanobacterial ecotypes

Association networks indicated that 11 of 14 cyanophage host genes were associated with particular cyanobacterial ecotypes after we took into account environmental variables (temperature, salinity, nitrate, phosphate; Fig. 8). In networks with positive associations only, and both positive and negative associations, the proportion of myo-cyanophage with nitrite assimilation gene *nirA* and purine synthesis gene *purS* are strongly positively statistically-associated with ODZ *Prochlorococcus* ecotype LLV, and purine synthesis genes *purC* and *purM* associated with LLVII *Prochlorococcus* (Fig. 8). Interestingly, when negative correlations were allowed, purine synthesis gene *purN* and pyrimidine synthesis gene *pyrE* negatively associated with HLII *Prochlorococcus,* rather than positively associating with a single LLII, LLIV and LLVII ecotype (Fig. 8B). The proportion of myo-cyanophage with *psbA* was weakly negatively associated with LLV *Prochlorococcus* (Fig. 8B). The associations seen in the network analysis are supported by simple correlations between ecotypes and the proportion of cyanophage with these viral host genes (Fig. S10). In particular, the proportion of cyanophage with many of the purine genes form the best correlation with a combination of the extremely low light groups (LLII, LLVII, LLV). This pattern could also be thought of as a negative correlation with High Light groups. LLI *Prochlorococcus* does not play a role in the networks, and none of the cyanophage host genes were specific to LLI infecting cyanophage.

## Nitrite assimilation in Prochlorococcus

Assimilatory nitrite reductase *nirA* and nitrite transporter *nirC* are variable genes in *Prochlorococcus*. Here we show a clear trend where the proportion of *Prochlorococcus* with the ability to use nitrite increases with depth (Figs. 3–6, Figs. S1–S4). *Prochlorococcus* in the mixed layer, do not have the transporter gene *nirC* (Figs. 3–6, Figs. S1–S4). A small fraction did have assimilatory nitrite reductase gene *nirA* in the mixed layer (∼20% on the GA03 transect and at HOT, ∼30% at GP13 stations and ∼10% in the Mediterranean) (Figs. 3–6, Figs. S1–S4). However, at all stations, ∼100% of *Prochlorococcus* contained both *nirA* and *nirC* at the base of the euphotic zone, and in the ODZs (Figs. 3–6, Figs. S1–S4). Nitrite data was not available for all of our stations. For the GA03 transect, where we have nitrite concentration data, we can see that *Prochlorococcus* has the ability to use nitrite at depths where nitrite is present (Figs. 2 and 3, Figs. S1–S2).

The addition of cyanophage *nirA* genes to the phylogenetic tree was critical. Previously, we examined *nirA* in LLV cyanobacteria in the ETNP ODZ and found ∼6 copies per cyanobacterial genome (*Widner et al., 2018*). In this reanalysis of the same dataset, where the tree now included viral *nirA*, the host copies of *nirA* were reduced to ∼1 copy per genome in the ETNP ODZ. Unidentified abundant viral host genes can skew results for bacterial analyses.

## Cyanophage cobS phylotypes

Since *terL* is too conserved to break into smaller phylotypes, we chose to examine *cobS* gene phylotypes to represent shifts in myo-cyanophage identity. The *cobS* gene was in the majority of myo-cyanophage, its proportion in myo-cyanophage generally did not increase or decrease with depth (Figs. 3–6, Figs. S1–S4), and the gene had 20 phylotypes (Fig. S11), making it a good choice to track the myo-cyanophage community. However, at the GP13 stations, the proportion of myo-cyanophage with *cobS* was abnormally low (Fig. 4, Fig. S3). Still, these *cobS* phylotype data give us a good idea about how myo-cyanophage shift over depth and space. At all stations, there was a shift in *cobS* phylotypes with depth (Fig. 9, Fig. S12). Several of the *cobS* phylotypes had no cultured representation (Fig. S11). Upper euphotic zone phylotypes at HOT, GA03, and GP13 include phylotypes 4, 6 (uncultured), and 8 (Fig. 9, Fig. S12). However, in the Mediterranean, the euphotic zone was dominated by cyanophage phylotypes 7 and 20 (Fig. 9). At all stations, the main phylotypes at depth were phylotypes 2 and 1 (uncultured). At some stations phylotype 2 dominated in the deep euphotic zone (HOT and GA03 St 3, 4, 6, 8, 10) and at other stations cyanophage phylotype 1 dominated the deep euphotic zone (GA03 St 14, 16, 18, 20, 22, 24) (Fig. 9, Fig. S12). Abundances of cyanophage *cobS* phylotypes can be seen in Table S5.

An RDA plot of sample similarity, as determined by *cobS* phylotypes, showed distinct trends in depth regime (mixed layer, chlorophyll maximum, deep euphotic zone, euphotic/mesopelagic transition), with additional separation by ocean basin (Fig. 10B). The ODZ samples were particularly distinct from the oxic samples (Fig. 10B). Statistical analysis indicated that the *cobS* community variability was partially explained ($p < 0.05$) by ocean basin (28%), depth (5%) and oxygen (2%).

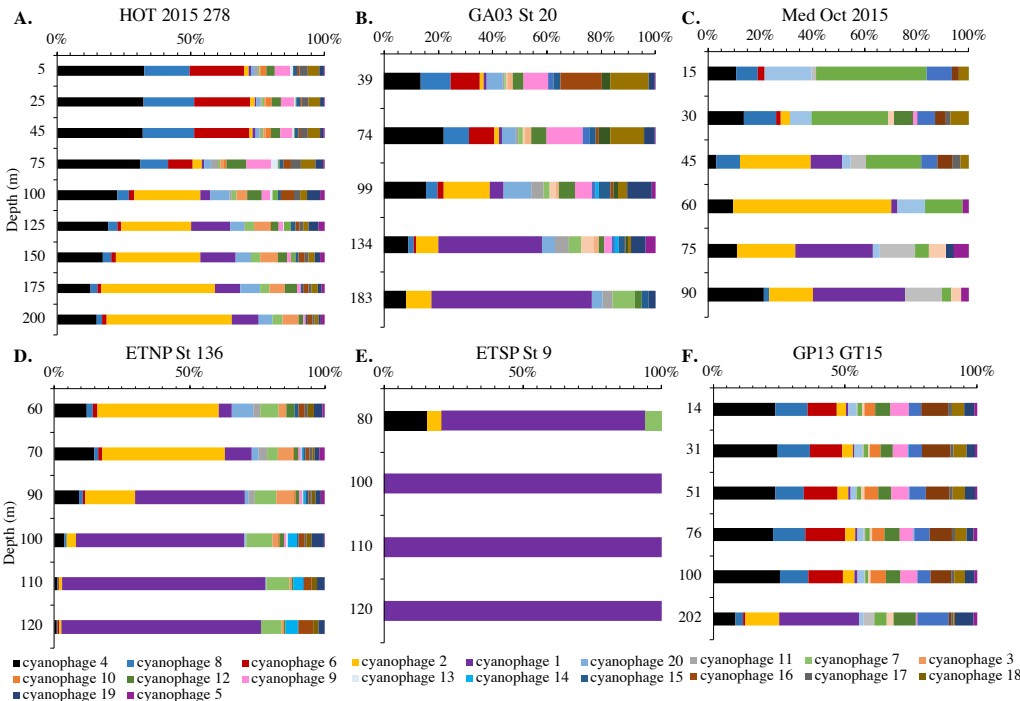

**Figure 9 Changes in myo-cyanophage community with depth as determined by *cobS* phylotype.** Myo-cyanophage core gene *cobS* phylotypes for (A) HOT 278 (Nov 2015), (B) North Atlantic GA03 St 20, (C) Mediterranean, (D) ETNP St 136, (E) ETSP St 9 and (F) South Pacific GP13 St GT15. *cobS* phylotypes were arranged by abundance. The labeled *cobS* phylogenetic tree can be seen in Fig. S11.

The myo-cyanophage community in the ODZs differed from those seen in oxic waters. The uncultured cyanophage phylotype 1 dominated the ODZ samples (Fig. 9). When the Simpson Index of Diversity (1-D) was calculated for myo-cyanophage *cobS* phylotypes, the oxic stations retained a diversity index >0.8 through the euphotic zone for most stations (Fig. 10, Fig. S13), though GA03 St 18 and St 20 did have a small decrease in diversity at their deepest depths (Fig. S13). However, the diversity of *cobS* phylotypes greatly decreased in anoxic ODZ waters, reaching zero in the ETSP (Fig. 10A). Thus, the ODZs appear to have a depauperate cyanophage community.

For comparison, we examined the Simpson Index of Diversity for cyanobacterial ITS ecotypes as well. The ETSP cyanobacteria also had very low diversity in the ODZ (Fig. S14). In the ETNP, the highest cyanobacteria diversity was in the oxycline (90 m), and diversity was reduced from that in the ODZ (Fig. S14). However, in general, the patterns were not as clear as for *cobS* because the cyanobacteria had very low diversity in surface waters, which were dominated by HLII (Fig. S14). However, HLII is a large group with many known sub-populations (*Kashtan et al., 2014*; *Kashtan et al., 2017*), so this low surface diversity may be due to operational definitions.

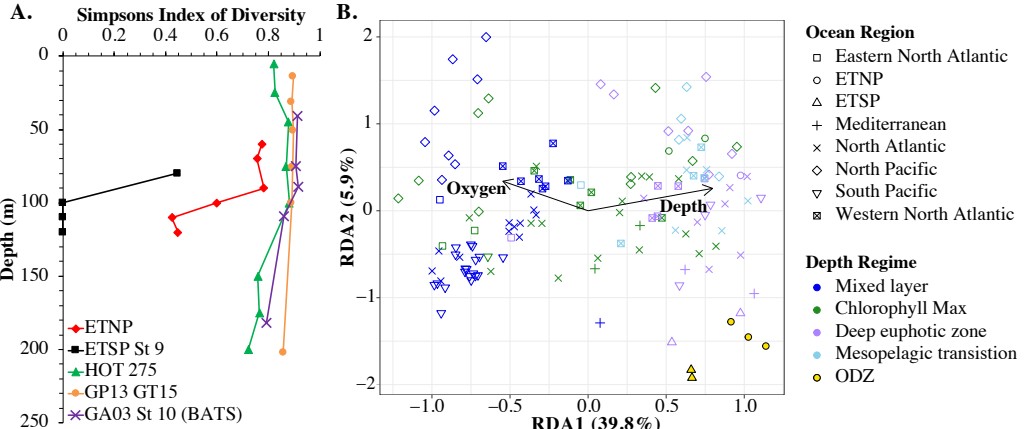

**Figure 10** **Myo-cyanophage community metrics calculated using *cobS* phylotypes.** (A) Simpson's Diversity Index (1-D) for *cobS* phylotypes of select stations. Other stations can be seen in Fig. S13. (B) An RDA plot of *cobS* phylotype data with ocean region shown by shape and depth regime shown by color. In the RDA analysis, community structure is a function of oxygen concentration, depth (treated as a quadratic function depth + depth$^2$) and ocean region, but only the arrows for oxygen and depth (pointing in the same direction as depth$^2$) are shown.

## DISCUSSION

*Prochlorococcus* ecotypes have distinct cascade of ecotype optima at different light levels (*Ahlgren, Rocap & Chisholm, 2006*; *Zinser et al., 2007*). However, very few studies of viruses in the environment have examined detailed depth profiles in the euphotic zone. Here we show distinct structuring of host ecotypes, cyanophage and cyanophage host genes with depth in the euphotic zone across ocean basins. While each station examined is unique, larger trends with depth can be observed. It is likely that these trends are truly in regards to light levels, as light is known to be the parameter structuring *Prochlorococcus* ecotype composition (*Rocap et al., 2003*; *Kettler et al., 2007*). However, we do not have light measurements for the majority of the stations examined here. Since light diminishes with depth in the euphotic zone, depth stands in for light level in this work.

### cobS as a marker for myo-cyanophage community structure

The single copy core gene terminase (*terL*) is too conserved to illuminate the myo-cyanophage community structure. However, the proportion of myo-cyanophage with *cobS* was constant with depth and near 1 (Figs. 3–6, Figs. S1–S4), suggesting that each myo-cyanophage harbors just one *cobS* gene. The phylogenetic tree splits *cobS* into 20 phylotypes (Fig. S11), and those *cobS* phylotypes shift with depth (Fig. 9). These *cobS* phylotypes derive from a single transfer event from the host, so the evolution of *cobS* should track the evolution of the viruses (Fig. S11; *Ignacio-Espinoza & Sullivan, 2012*; *Fuchsman et al., 2021*). Therefore, we use *cobS* phylotypes to track changes in the myo-cyanophage community. Our *cobS* data implies that along with their hosts, the myo-cyanophage themselves shift identity with depth (Figs. 9 and 10B) and between ocean basins (Fig. 10B). For example, the Mediterranean, known for its high salinity (*Vargas-Yáñez et al., 2017*) and extremely low

phosphate (*Motegi et al., 2015*), had a unique assemblage of myo-cyanophage in the upper euphotic zone (Fig. 9). The Simpson Index of Diversity (1-D) for *cobS* phylotypes indicated that the majority of oxic stations retained high diversity (>0.8) throughout the euphotic zone (Fig. 10A, Fig. S13), but the ODZs were dominated by a single *cobS* phylotype (Fig. 9). Thus, the diversity of *cobS* phylotypes greatly decreased in anoxic waters (Fig. 10A). A similar decrease in diversity of myo-cyanophage host genes (*talC*, *thyX*, *psbA*) from oxic waters above the ETNP ODZ to anoxic waters was seen previously (*Fuchsman et al., 2021*). Here we show that this consistent decrease in *cobS* diversity is unique to anoxic waters and does not occur at the bottom of the oxic euphotic zone. Using cyanobacterial ecotypes, we also see a decrease in diversity in the ETSP ODZ, and a diversity decreased between oxic and anoxic waters in the ETNP (Fig. S14). However, there is only a small difference between the ETNP ODZ and deep oxic euphotic waters (Fig. S14). The decrease in myo-cyanophage diversity in the ODZs may be linked to reduced diversity of the *Prochlorococcus* host, but it would be better to analyze this including sub-ecotype scale diversity.

## Cyanophage host genes stratify vertically

The proportion of cyanophage with purine synthesis genes *purN*, *purM*, and *purC* and pyrimidine synthesis gene *pyrE*, as measured by viral host gene / *terL* ratios, increased with depth throughout the ocean (Figs. 3–6, Figs. S1–S4). Contrastingly, we find that generally the proportion of podo-cyanophage with *psbA* and myo-cyanophage with *psbD* were ~1 or greater in surface waters but were reduced with depth (Figs. 3–6, Figs. S1–S4). These trends appear to be conserved between ocean regions and over time. The proportion of cyanophage with these host genes depends on the ecotype of the host.

Network analysis indicates that 11 of the 14 cyanophage host genes are associated statistically with cyanobacterial ecotypes. These statistical links are present even when links to environmental factors are considered. Linear regressions indicate that the proportion of myo-cyanophage with *purN*, *purM*, *purC*, and *pyrE* genes are correlated to the combination of extremely Low Light *Prochlorococcus* ecotypes Low Light II, LLVII, and LLV *Prochlorococcus* (Fig. S10). This same correlation is seen in the network analysis as negative statistical associations with HLII *Prochlorococcus* (Fig. 8). On the other hand, *psbD* is statistically associated with High Light *Prochlorococcus* ecotypes (Fig. 8, Fig. S10). The HLI ecotype was seen in abundance in the colder South Pacific GP13 (Fig. 4, Fig. S3) and the Mediterranean (Fig. 5). However, the switch between HLII and HLI ecotypes did not seem to affect the profiles of cyanophage host genes examined here, except that cyanophage *psbD* was found at 2 copies per myo-cyanophage when HLI was present (Figs. 4–5).

It is interesting that none of the cyanophage host genes examined here correlate with LLI *Prochlorococcus*. High Light *Prochlorococcus* ecotypes have genes to deal with high light and low nutrients (*Kettler et al., 2007*) and live in conditions where these genes are needed. Extremely low light ecotypes (LLII, LLIV) do not have genes to deal with high light but rather have genes to deal with reduced light (*Kettler et al., 2007*), which is consistent with where they live in the water column. These ecotypes also grow slowly. Oxic *Prochlorococcus* living at 100–150 m depths in the Pacific divide approximately once per week while *Prochlorococcus* in the mixed layer divide once per day (*Vaulot et al., 1995*). On the other

hand, LLI *Prochlorococcus* also have genes to deal with high light (*Kettler et al., 2007*), but live in an ideal spot with available nutrients and a medium amount of light. The cyanophage infecting LLI may not need lineage specific viral host genes.

The requirements for a viral host gene, in principle, can inform about the limitations for viral infection for a given host. Requirements for particular nucleotide synthesis genes in viral infection of slow-growing, extremely low light ecotypes of *Prochlorococcus* indicate that the necessity for viruses to maintain particular elements of nucleotide production. In our previous work in the ETNP, we found that purine synthesis, but not pyrimidine synthesis genes increased with depth (*Fuchsman et al., 2021*). Latent period scales with host growth rates (*Nabergoj, Modic & Podgornik, 2018*). Therefore, we hypothesized that proteins for purine synthesis genes were needed by viruses because of elongated latent periods at depth (*Fuchsman et al., 2021*). Purines are used in energy storage and transfer (ATP), signaling (cyclic-AMP), and in cofactors (NADH) (*Zhao et al., 2013*). Thus, the cyanophage may need more energy and storage molecules created by the host due to their long latent periods at depth. However, outside of the ETNP ODZ, we also see the proportion of myo-cyanophage with pyrimidine synthesis gene *pyrE* increase with depth, including in the ETSP ODZ (Figs. 3–6, Figs. S1–S4). This indicates that purines and pyrimidines may also be needed as nucleotides to form viral DNA in the deep euphotic zone.

Some cyanophage host genes were present or absent specifically in the ODZ where the Low Light V ecotype of *Prochlorococcus* is found. The proportion of myo-cyanophage with the gene for Photosystem II D1 protein (*psbA*) is ∼1 but does not reduce with depth except in the ETNP and ETSP ODZs (Figs. 3–6, Figs. S1–S4). Myo *psbA* is weakly negatively statistically associated with Low Light V (Fig. 8). It was hypothesized that this reduction in myo *psbA* was due to the stable conditions in the ODZ, where high light levels never occur (*Fuchsman et al., 2021*). The myo-cyanophage purine synthesis gene *purS*, nitrite assimilation gene *nirA* and nitrite transporter *nirC* were also specifically found in high abundance in the ODZs and are positively statistically associated with Low Light V *Prochlorococcus* (Fig. 8). Purine synthesis genes are common in cyanophage genomes in the deep euphotic zone, so it is unclear why *purS* would only be abundant in the ODZs. However, we note that the diversity of myo-cyanophage in the ODZs is extremely low (Fig. 9). The dominant phage has a set of genes that are therefore dominant. However, these genes must still be useful, or a closely related competitor without the genes would take over.

Mixed layer depth differences across and between ocean basins appeared to be an important driver of ocean basin differences in cyanobacterial ecotype zonation through the water column. In the BioGeotraces GA03 transect, there were several stations in section 1 of the transect, which crossed the Gulf Stream, that had deep mixed layers (St 4, St 6, St 8) (Fig. 2) with chlorophyll extending only a little below the mixed layer. This area likely has deep mixed layers often because of the Gulf Stream. At these section 1 stations, similarly to other stations examined here, the proportion of myo-cyanophage that had *psbD* was reduced below the mixed layer and the proportion of myo-cyanophage with *purN, purM,* and *purC* increased below the mixed layer. However, the proportion of podo-cyanophage with *psbA* was constant with depth, instead of decreasing. At stations in section 2 of the GA03 transect

and at HOT where the mixed layer was shallower, the proportion of podo-cyanophage with *psbA* decreased with depth. Thus, the proportion of podo-cyanophage with *psbA* did not have as consistent depth profiles as other host genes, at least at stations with deep mixed layers. Viral *psbA* is used to replace the photosynthesis gene most vulnerable to light degradation (*Lindell et al., 2004*). It is possible that the consequence of not having *psbA* in a mixed water column is larger than the consequences of having extra genes for the purine pathway.

## Cyanophage encode nitrite assimilation genes in ODZs

Most Low Light *Prochlorococcus* sequenced strains have the *nirA* gene for assimilatory nitrite reductase and some High Light strains have *nirA* (*Berube et al., 2019*). The gene for nitrite transporter *nirC* follows similar patterns in *Prochlorococcus*. In our dataset, the fraction of *Prochlorococcus* able to use nitrite increased with depth; in the mixed layer 10–30% of *Prochlorococcus* genomes contained *nirA* and ~0% contained *nirC*, but at the base of the euphotic zone ~100% of *Prochlorococcus* contained both genes (Figs. 3–4, Figs. S1–S4). The Low Light I ecotype of *Prochlorococcus* are generally coincident with primary nitrite maximum at the base of the euphotic zone (Fig. 2), which could explain their ability to assimilate nitrite.

We note that though nitrite assimilation genes were found in all *Prochlorococcus* cells at depth in oxic waters, viruses containing *nirA* or *nirC* were not detectable in oxic metagenomes (Figs. 3–4, 6, Figs. S1–S4). However, both *nirA* and *nirC* genes were in 100% of myo-cyanophage in ODZ waters and though no known podo-cyanophage have *nirA,* high cyanophage *nirA* numbers indicate either multiple copies of the gene in myo-cyanophage or that *nirA* may also be in podo-cyanophage (Fig. 6). Host nitrogen assimilation can be especially important to viruses because viruses are enriched in N and P compared to microorganisms (*Jover et al., 2014*). Low environmental N concentrations can cause viral burst size to decrease significantly (*Cheng, Labavitch & VanderGheynst, 2015*). The presence of nitrite assimilation genes in cyanophage highlights the particular importance of nitrite as an N source to *Prochlorococcus* in ODZ waters. Gene analysis of Low Light V *Prochlorococcus* in ODZs indicated that they have the capability of utilizing ammonium, urea, nitrite, and nitrate (*Astorga-Elo et al., 2015*; *Widner et al., 2018*; *Ulloa et al., 2021*). Addition experiments indicate that the *Prochlorococcus* in ODZs would prefer to use ammonium (*Aldunate et al., 2020*), but in the ODZs, ammonium is usually below detection (<10 nM in the ETNP; *Widner et al., 2018*) (<60 nM in the ETSP; *Widner, Mordy & Mulholland, 2018*). Additionally, *Prochlorococcus* must compete with anammox bacteria and nitrite oxidizers for this ammonium (*Penn et al., 2019*) and also for other reduced N forms such as urea, also below detection in the ODZ (*Ganesh et al., 2018*; *Widner et al., 2018*). On the other hand, nitrite concentration can reach several micromolar (Fig. 6). *Prochlorococcus* assimilatory nitrite reductase protein NirA was found in proteomic analyses in the ETNP ODZ, indicating activity (*Fuchsman et al., 2019*), and *in situ* stable isotopes indicate that Low Light V *Prochlorococcus* predominantly utilize nitrite, rather than ammonium, in the ODZ (*Aldunate et al., 2020*). So, while the partially oxidized N in nitrite

may not be preferred, it is an available and utilized N source for ODZ *Prochlorococcus* and is important enough for nitrite assimilation genes to be in ODZ cyanophage.

## Cyanophage encode pstS when phosphate is scarce

Similar to data in *Kelly et al. (2013)*, the proportion of myo-cyanophage with the phosphate transporter, phosphate binding protein gene (*pstS*) varied by ocean basin with high abundances in the North Atlantic and Mediterranean and low abundances in the South Pacific, North Pacific, and ODZs. The North Atlantic and Mediterranean are known to have extremely low phosphate concentrations in the euphotic zone (*Wu et al., 2000*; *Kelly et al., 2013*; *Motegi et al., 2015*). The GA03 transect extended across a region of the North Atlantic generally found to have undetectable phosphate according to World Ocean Atlas (*Garcia et al., 2019*) (Fig. 1). The Mediterranean station sampled also had extremely low phosphate concentrations (Fig. 7) (*Haro-Moreno et al., 2018*). At HOT, phosphate concentrations are often <100 nM, but much of the community uses dissolved organic P (*Björkman, Thomson-Bulldis & Karl, 2000*). Phosphate concentrations are >100 nM in the South Pacific, and those phosphate concentrations are large enough to allow organisms there to make phospholipids without needing lipid renovations (*Van Mooy et al., 2009*). Cyanophage did not carry the *pstS* gene at the South Pacific stations (Figs. 4 and 7, Fig. S3). Interestingly, the proportion of cyanophage with *pstS* at a particular station did not appear to change with depth in the oxic ocean. The cyanophages in regions that typically have extremely low phosphate appear to retain *pstS* even when phosphate was measurable at a particular depth at the time of sampling (Fig. 7). These cyanophages seem adapted for potential future phosphate limitation. The ETNP is an exception because the proportion of cyanophage with *pstS* decreases precipitately when switching from oxic to anoxic conditions (Fig. 6) (*Fuchsman et al., 2021*). In the ETNP, the phosphate concentrations increased dramatically from 0.3 µM in surface oxic waters to reaching 3 µM in the ODZ (*Fuchsman et al., 2021*). However, the phosphate concentrations in the ETNP surface waters still were larger than found at the other stations examined here. In any case, the proportion of myo-cyanophage with *pstS* follow the phosphate climatology for the region rather than exact concentrations.

We hypothesized that cyanophage heme oxygenase (*pbsA*) would have similar proportional abundance patterns related to iron availability because heme oxygenase has been associated with iron acquisition in some bacteria (*Frankenberg-Dinkel, 2004*). The highest proportions of myo-cyanophage with the heme oxygenase gene were at the South Pacific GP13 stations (Figs. 3–6, Figs. S1–S4). From measured iron and nutrient concentrations, these GP13 stations are hypothesized to be iron limited (*Ellwood et al., 2018*). However, differences in proportions of myo-cyanophage with heme oxidase between stations and regions in this dataset were not large. In cyanobacteria, heme oxygenase is used in phytobilin biosynthesis, even in *Prochlorococcus*, which use divinyl chlorophylls for light harvesting (*Frankenberg-Dinkel, 2004*). The role of cyanophage heme oxygenase has also been shown to be phytobilin biosynthesis (*Dammeyer et al., 2008*). Thus, the factors affecting the proportion of myo-cyanophage with heme oxidase are unclear.

## CONCLUSIONS

We find consistent patterns of key cyanophage host genes across the oxic ocean. Cellular metagenomes from the oxic ocean are dominated by myo-cyanophage, except in the hypoxic waters above ODZs and in the Mediterranean. The majority of these cyanophage host genes correlate/statistically associate with host ecotype. However, *Prochlorococcus* ecotype abundances are known to closely follow the environmental parameters light, temperature, and oxygen concentrations (*Rocap et al., 2003*; *Ahlgren, Rocap & Chisholm, 2006*; *Zinser et al., 2007*; *Lavin et al., 2010*; *Malmstrom et al., 2010*). Cyanophage in surface waters have host genes to deal with high light, and cyanophage in deep waters have host genes to produce nucleotides. LLI *Prochlorococcus*, which live at depths with medium light and available nutrients, do not have particular cyanophage host genes associated with them. Cyanophage phosphate transporter gene *pstS* was the exception to these trends, with high abundance in ocean basins with extremely low phosphate concentrations rather than being linked to a *Prochlorococcus* ecotype. Thus, we hypothesize that abundances of cyanophage host genes related to nutrient acquisition may diverge from host ecotype constraints because nutrient concentrations can vary under the same light, temperature, and oxygen conditions.

The Simpson Diversity Index (1-D) indicated that the myo-cyanophage in the anoxic ODZ are a depauperate community. The *Prochlorococcus* host ecotypes also decreased in diversity over the oxic/anoxic transition, but results were not as clear due to the dominance of HLII in surface waters. By comparison to the oxic ocean, we can see which cyanophage host genes are especially important (*nirA, nirC,* and *purS*) or unimportant (*psbA* and *pstS*) in Oxygen Deficient Zones. The importance/unimportance of these cyanophage host genes highlights both the stability of conditions in the ODZ and the importance of nitrite as an N source to LLV *Prochlorococcus* in the ODZ.

## ACKNOWLEDGEMENTS

We thank Michael C.G. Carlson for helpful advice on cyanophage analyses, and J.L. Weissman for guidance on network analysis. The GEOTRACES 2021 Intermediate Data Product (IDP2021) represents an international collaboration and is endorsed by the Scientific Committee on Oceanic Research (SCOR). We thank the researchers involved in Geotraces and the Hawaii Ocean Timeseries for their data collection and quality control. We thank Michael Ellwood for providing CTD fluorescence data for the GP13 stations, which are not in IDP2021.

### Funding

This work was funded by start-up funding from Horn Point Laboratory. The funders had no role in study design, data collection and analysis, decision to publish, or preparation of the manuscript.

## Grant Disclosures

The following grant information was disclosed by the authors:
Horn Point Laboratory.

## Competing Interests

The authors declare there are no competing interests.

## Author Contributions

- Clara A. Fuchsman conceived and designed the experiments, performed the experiments, analyzed the data, prepared figures and/or tables, authored or reviewed drafts of the article, and approved the final draft.
- David Garcia Prieto performed the experiments, analyzed the data, prepared figures and/or tables, authored or reviewed drafts of the article, and approved the final draft.
- Matthew D. Hays performed the experiments, analyzed the data, authored or reviewed drafts of the article, and approved the final draft.
- Jacob A. Cram conceived and designed the experiments, performed the experiments, analyzed the data, prepared figures and/or tables, authored or reviewed drafts of the article, and approved the final draft.

## Data Availability

The code for network analysis is available at Figshare: Cram, Jacob (2022): CyanoVirLasoo. figshare. Software. https://doi.org/10.6084/m9.figshare.21498402.v1.

The data (abundances in publicly available metagenomes) is available in the Supplemental Tables.

The reference sequences used for these trees can be found at Figshare: https://doi.org/10.6084/m9.figshare.21899361.v1.

## Supplemental Information

Supplemental information for this article can be found online at http://dx.doi.org/10.7717/peerj.14924#supplemental-information.

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
