# Peer review of "Associations between picocyanobacterial ecotypes and cyanophage host genes across ocean basins and depth"

_PeerJ, doi:10.7717/peerj.14924_

## Round 0.1 · original submission · Minor Revisions

Looking forward to your point-by-point reply to all of the reviewers' comments.

Reviewer 1 ·

Basic reporting

, the manuscript is interesting and rigorous but it is a very lengthy article whose readability could be improved by removing part of the description of the metagenomes and focusing more on the phage genes and their association with host ecotype and actual environment.

Experimental design

It is appropriate

Validity of the findings

No comment

Additional comments

This is an interesting and well-carried-out work that approaches a hot topic of virus-host interactions: the contribution of host metabolic genes (formerly AMGs). This has been carried out analyzing depth profile metagenomes of which there is already a significant bulk. The authors describe a correlation between the metabolic genes found in the viral genomes and the environment and host ecotype. They have used an approach called "phylogenetic metagenomic read placement" which they claim is better than just sequence comparison. They have used assembled or non-assembled reads depending on the gene or ITS. In general, the manuscript contains much valuable data and carries an important message applicable cyanophages and potentially to viruses in general which is how finely tuned they are to the ecology of the host. My main criticism (that the authors may choose to ignore) is that the description (particularly the first part of the results not dealing with phage genes) is too lengthy and comprehensive. It might be more productive in terms of the impact of the paper if it is kept shorter and more focused.

Ln 57 to 62 are not really needed, the intro is already very long and some parts sound repetitive, it would improve the manuscript more focused writing.
Ln 63 “host function” is classically known as auxiliary metabolic genes (AMGs), right?
Ln 314 to 415 seem part of a separate manuscript, consider removing them altogether or just summarizing the main findings.
Ln331 I miss a description of the season and temperature gradient like in the GA03 transect
Ln 476 whole paragraph. This part is a bit confusing pstS does not change with depth (when there is no oxygen minimum zone) but it does when it is present. Why is it more related to nutrient concentrations than to host ecotype? This requires elaboration not only references to Fuchsman et al, 2021.
Ln 600 The Mediterranean is also known to be limited by P more so than other water masses
Ln 609 Viruses do require the host to produce ATP (e.g. respire) for replication during the lytic cycle
Ln303 the proportions of myo and podo are different in the metagenome than in the metavirome so I would specify that were more abundant “in the metagenomes”
Ln 612 likely the second
Ln755 in surface

Reviewer 2 ·

Basic reporting

No comment.

Experimental design

No comment.

Validity of the findings

No comment.

Additional comments

The paper is very well analyzed, and then discussed the distribution of cyanobacterial ecotypes and viruses and host-like genes and their significance.

I have only a simple question, why is the siphovirus not covered?

Minor coments.
L208: The format of the citation is wrong. Please remove the Parentheses throughout the text.
L755: typo. Cyanophage in surface water.

Reviewer 3 ·

Basic reporting

no comment

Experimental design

Provide fasta files in figshare for all genes used to generate trees in order to facilitate the ability of other researchers to replicate your analysis.

Provide version numbers and parameter flags for all software used in the analysis.

Validity of the findings

Lines 450-451 "The cyanophage host genes that increase with depth must only provide benefit for viral production at depth."
Lines 457-458. "The cyanophage host genes that decrease with depth must only provide benefit for viral production in surface waters."
There is good evidence in the manuscript that these genes are selected relative to depth (though this might not be the proximal selective pressure and the authors do state that they are using depth as a proxy for light). But, I think it goes a bit far to state that these genes "must only provide benefit for viral production."

Regarding lines 764-767: "The Simpson Diversity Index (1-D) indicated that the myo-cyanophage in the anoxic ODZ are a depauperate community, implying either that lack of oxygen is a barrier for cyanophage despite their use of their host as an energy source or that the Prochlorococcus host could be less diverse in the ODZ." I'm not inclined to agree that there's ample evidence for the the oxygen hypothesis. Are there data from the literature that link low O2 to decreased rate of virus production? Also, the authors have sufficient data to link Prochlorococcus diversity to cyanophage diversity (using either ITS or an alternative Prochlorococcus marker gene). The authors are missing supporting results that could buttress their hypotheses on the lack of diversity in the anoxic ODZ cyanophage community.

Additional comments

Lines 90-92 "Therefore, the presence of a viral host gene suggests that a process is beneficial to the survival of a microbial viral factory between the time of infection and lysis."

It would be useful to express this within the framework of natural selection.


Lines 128-129 "Distinct ecotypes of Prochlorococcus (LLV, LLVI, LLVII) live in Oxygen Deficient Zones (ODZs)"

The clade names need to be fixed here and elsewhere. LLVII is the same as NC1 (see: https://doi.org/10.1038/nrmicro3378). Note, however, that this group is not monophyletic and there's limited evidence to identify it as a distinct ecotype. The Ulloa et al. paper (https://doi.org/10.1073/pnas.2025638118) also renames the LLV and LLVI clades to AMZ I and AMZ II, respectively, and further identified the AMZ III clade. Did the authors consider looking at AMZ III in their analysis? Regarding the ODZ clade names, maybe it would be best to provide both the ones from the Lavin paper and the Ulloa paper to limit confusion.


Figure 10. Which scaling is being shown for the RDA plot? Please provide the proportion of variability explained by each axis in the figure plot.


Lines 227-228. "Very few reads were removed at this step."

Be more specific. It would be sufficient to provide a range of percentages instead of for each metagenome/gene.


Lines 499-503 "In the BioGeotraces GA03 transect, there were several stations in section 1 of the transect that had deep mixed layers (St 4, St 6. St 8) (Figure 2). At section 1 stations with deep mixed layers, the proportion of myo-cyanophage that had psbD was still reduced at depth and the proportion of myo-cyanophage with purN, purM, and purC increased with depth, similarly to other stations examined here."

Looks like St 4, St 6. St 8 on GA03 with deep mixed layers are in the gulf stream current. The GP13 samples were taken in the winter when physical processes (e.g. wind and temperature) can drive mixing. I think the more important observation here is that the purN, purM, purC, and psbD genes didn't show consistent proportions with depth in deeply mixed water columns. Does selection happen faster for the purN, purM, purC, and psbD genes in particular, i.e., on time scales relevant to movement of water between the surface and at depth within mixed layers? Some commentary on this topic is needed here.

---

## Round 0.2 · accepted · Accept

Thank you for the thorough revision of your paper, which now can be accepted by PeerJ.

Reviewer 1 ·

Basic reporting

The paper is now ready for publication

Experimental design

The paper is now ready for publication

Validity of the findings

The paper is now ready for publication

Additional comments

The paper is now ready for publication